# Human Microbiota Network: Unveiling Potential Crosstalk between the Different Microbiota Ecosystems and Their Role in Health and Disease

**DOI:** 10.3390/nu13092905

**Published:** 2021-08-24

**Authors:** Jose E. Martínez, Augusto Vargas, Tania Pérez-Sánchez, Ignacio J. Encío, Miriam Cabello-Olmo, Miguel Barajas

**Affiliations:** Biochemistry Area, Department of Health Science, Public University of Navarre, 31008 Pamplona, Spain; martinez.137785@e.unavarra.es (J.E.M.); augusto.vargas@unavarra.es (A.V.); tania.perez@unavarra.es (T.P.-S.); ignacio.encio@unavarra.es (I.J.E.)

**Keywords:** microbiota, dysbiosis, crosstalk, metabolites, human diseases

## Abstract

The human body is host to a large number of microorganisms which conform the human microbiota, that is known to play an important role in health and disease. Although most of the microorganisms that coexist with us are located in the gut, microbial cells present in other locations (like skin, respiratory tract, genitourinary tract, and the vaginal zone in women) also play a significant role regulating host health. The fact that there are different kinds of microbiota in different body areas does not mean they are independent. It is plausible that connection exist, and different studies have shown that the microbiota present in different zones of the human body has the capability of communicating through secondary metabolites. In this sense, dysbiosis in one body compartment may negatively affect distal areas and contribute to the development of diseases. Accordingly, it could be hypothesized that the whole set of microbial cells that inhabit the human body form a system, and the dialogue between the different host microbiotas may be a contributing factor for the susceptibility to developing diseased states. For this reason, the present review aims to integrate the available literature on the relationship between the different human microbiotas and understand how changes in the microbiota in one body region can influence other microbiota communities in a bidirectional process. The findings suggest that the different microbiotas may act in a coordinated way to decisively influence human well-being. This new integrative paradigm opens new insights in the microbiota field of research and its relationship with human health that should be taken into account in future studies.

## 1. Introduction

Evolution has been involved in the development of all microorganisms, and during this evolutionary process, many of them have co-evolved with humans, inhabiting different parts of the body and forming what is known as human microbiota [1]. Lederberg and McCray first introduced the concept of “microbiota” in 2001, referring to it as “the ecological community of commensal, symbiotic and pathogenic microorganisms that literally share our body space and have been all but ignored determinants of health and disease” [2]. Although they are predominantly anaerobic bacteria [3], we can also find viruses, fungi, archaea, and even protists [4]. According to the literature, the microbiota has a ratio of about 1:1 when compared to the number of human cells, meaning that a reference person hosts about 4 × 10^13^ bacteria [5,6]. Interestingly, most of these microorganisms are found in the digestive tract, which harbors between 150 and 400 different bacteria species [7]. Firmicutes and Bacteroidetes are the most predominant phylum followed by Actinobacteria, Proteobacteria, and Synergistetes [8,9].

In addition, the microbiota presents a reservoir of genes (microbiome) that is way larger than the human genome [10,11]. Specifically, the colon is the most colonized section with a mass of microorganisms close to 1.5 kg of weight [12]. Nevertheless, this is not the only area where microorganisms are hosted, as they also inhabit the skin, the vagina, the genitourinary tract, the respiratory tract, and the oral cavity [13].

Besides the great controversy around the topic, most scientific works support the hypothesis that humans are born sterile and have their first contact with exogenous microorganisms at the time of birth. Depending on the type of delivery, the newborn has the first contact with their mother’s birth canal (vaginal microbiota) or belly skin (skin microbiota) in the case of caesarean deliveries [14,15]. Besides the process of colonization lasting approximately three years [16], the type of delivery has a profound impact on the child’s microbiota colonization and immune system programming, and influences the risk of suffering many diseases in adulthood [17,18,19]. Besides the delivery mode, other factors such as breastfeeding [17] and adulthood diet [20], exposition to agents such as drugs [21] or antibiotics [22,23], physical activity [24,25], psychological stress [26], sleep quality [27], and host’s circadian clock [28], among others, collectively contribute to the microbiota composition and activity.

The microorganisms that currently conform the mammalian microbiota have gone through selection pressure, and they survived due to the advantageous role they play in the host homeostasis [29]. As mentioned above, an enormous diversity of bacteria have been found in the different body compartments [30], and this specific bacteria species create a particular ecosystem whose equilibrium importantly affects the proper functioning of this body compartment. This compartmentalization may be explained by the divergent characteristics of each organ and tissue [31]. Most of this knowledge comes from large research projects, and special attention must be paid to the Human Microbiome Project (HMP), an ambitious project aimed to make it visible the power of the microbiota in health and disease [32]. During the first phase (HMP1), 250 subjects were sampled and screened for their microbial genomes in five body sites which are considered significant: the mouth, the nasal cavity, the vagina, the gastrointestinal track and the skin (for details see [32]). The HMP1 contributed with invaluable findings on the subject and led to several publications [33,34], as the same time that new gaps were identified. During the second phase, The Integrative Human Microbiome Project (HMP2), which specifically focused on some physiological conditions (pregnancy, irritable bowel syndrome, and prediabetic state), efforts were coordinated to explore the function of the intestinal microbes [35]. The combination of metagenomics with other -omic platforms (such as metabolomics, metaproteomics, and metabolomics) will provide an overview of the current activity of the microbiota [36,37], the HMP2 highlighted interesting findings on the interplay between host and its resident gut microorganisms [35].

As any biological system, the microbiota is dynamic and fluctuates. A healthy (eubiotic) microbiota is resilient and can restore equilibrium when it undergoes oscillations [38,39]. On the contrary, in other cases, host and environmental factors restrain the microbiota from compensating the alterations, and dysbiosis occurs. This is important since dysbiosis has been related to multiple pathologies in humans, including metabolic disorders such as diabetes mellitus [40], obesity [41], and non-alcoholic fatty liver disease (NAFLD) [42], inflammatory diseases like inflammatory bowel disease or asthma [43], or cognitive dysfunctions like Alzheimer’s disease [44] and autism spectrum disorder [45]. Good evidence for this can be found in the list of ongoing research projects within the Horizon 2020 program on the role of microbiota in specific conditions such as cardiometabolic diseases (MetaCardis), chronic inflammatory diseases (SISCID), cancer (GOMS), chronic liver disease (MICROB-PREDICT), or autism (Gemma).

The main objective of this article is to summarize the available evidence on the possible communications between the different microbiota niches in the human body. Besides, we will also refer to interactions between microbiota communities and different organs. In the following sections, we will go deeper into the main commensal microbiota communities that shape the human body, define the dominant microorganisms that reside in them, and establish possible connections between the different microbiota niches and between these microbiotas and specific diseases or disorders. We focused on the major body sites investigated within the HMP1 (skin, oral, vaginal, and gut) [32]. In addition, for a more inclusive understanding, we decided to include the respiratory tract microbiota, that has recently attracted much attention in the current situation with the COVID-19 pandemic disease [46,47], as well as the urinary [48] and the penile microbiota [49].

## 2. Main Microbial Communities in the Human Body

### 2.1. Skin Microbiota

The skin is a complex organ which provides the first mechanical and biological barrier between the environment and the human cells. It is divided into two main layers: the epidermis and the dermis. Most bacteria species in the skin microbiota belong to Actinobacteria, Firmicutes, and Proteobacteria phyla [50], and four main genera, *Corynebacterium*, *Propionibacterium*, *Staphylococcus*, and *Streptococcus* [50,51].

They are located according to their environmental requirements so that anaerobic microorganisms like *Propionibacterium* spp. are placed in sites with more anaerobic conditions such as the sebaceous glands, and other more tolerant bacteria, like *Corynebacterium* spp., are distributed along the whole skin site [51]. Besides it is one of the largest organs of the human body, the skin ranked fourth place in the human body’s part with the highest number of bacteria [50,52].

In utero, the baby’s skin is sterile, therefore the skin microbiota is established few moments after birth. From this point, the microbes colonize the skin until it reaches an equilibrium [53]. The type of delivery is crucial for the configuration of the skin microbiota in the baby. In this way, children born in a natural way (birth canal) present bacterial communities similar to that in the mother’s vaginal microbiota, mainly *Lactobacillus* and *Prevotella* spp. On the other hand, those children born by C-section have microorganisms from the mother’s skin microbiota, predominantly *Propiniobacterium*, *Corynebacterium*, and *Staphylococcus* [15].

Interestingly, the distribution of the microbial communities on the skin surface is not homogeneous. One study reported that the front part of the body is more colonized, and is represented by *Propionibacterium*, *Corynebacterium*, and *Proteobacteria*, while the rear is represented by *Staphylococcus*, *Corynebacterium*, and *Propionibacterium* [50]. Data indicates that differences exist between female and male skin microbiota. Such discrepancies are driven by important dissimilarities in factors such as hormone production, sebum production or make-up use, that dramatically influence the environmental conditions of the skin [54]. Besides sex, aspects such as personal hygiene, immune status or the presence of skin diseases influence the structural composition of the skin microbiota [50]. In addition, another study indicated that environmental factors such as lifestyle can also impact skin microbiota since significant differences were identified between US residents, that live a Western lifestyle, and Amerindians from the Amazonas [55].

Alterations in the skin microbiota are related to some skin disorders. To illustrate, acne is triggered by bacteria overgrowth of *Propionibacterium* spp. and particularly *Propionibacterium acnes* [50,56]. In addition, *S. aureus* spp. (*S. aureus* and *S. epidermidis*) and *Malassezia* spp. fungi were identified in most cases of atopic dermatitis [51,57], and some *Corynebacterium* spp. were related to the onset of AD [58]. Besides, people with hidradenitis suppurativa, an inflammatory skin disease, present an enrichment in *Corynebacterium*, *Porphyromonas*, and *Peptoniphillus* spps. [59]. Bacteria are not the only microbial cells that can promote skin problems, since a mite (Demodex) and a fungi (Malassezia) were shown to be involved in the development of Rosacea and Seborrheic dermatitis, respectively [50].

### 2.2. Oral Microbiota

The oral microbiota is an important part of the human microbiota and has been described to harbor more than 700 different microbial species. The fact that it is close to many other anatomic regions makes the oral microbiota the second most complex microbiota niche in the human body after the gut [60,61]. There are discrepancies between studies regarding the main component of the oral microbiota, and while some authors suggest that the most important phyla in the oral microbiota are Actinobacteria, Bacteroidetes, Firmicutes, Proteobacterias, and Synergistetes, others pointed to other phyla such as Fusobacterias and Spirochaetes [62].

Within the oral microbiota, certain species like Fusobacterium, Gemmela, Veillonella, Streptococcus, or Granulicatella are ubiquitous, while others like Bacteroidetes, Pasterutella, Prevotella, Neisseria, and Corynebacterium are associated to some particular regions [63].

The oral cavity comprises many different surfaces including saliva, soft tissues (cheek, palate, and tongue), and hard tissues (tooth), where bacteria and other microorganisms could potentially colonize and predominate [60]. For instance, saliva is predominated by *Streptococcus*, *Veillonella*, and *Prevotella*, meanwhile, the surface of soft tissues are colonized by *Streptococcus salivarus*, *Rothia*, and *Eubacterium.* The teeth are also home to microorganisms. Members from the *Corynebacterium*, and *Actinomyces* genera normally colonize the supragingival region, while the subgingival area is characterized by anaerobic species from the *Spirochaetes*, *Fusobacteria*, *Actinobacteria*, *Proteobacteria*, and *Bacteroidetes*, genera [60].

Oral dysbiosis has been related to certain diseases. For example, oral candidiasis, that was linked to caries, is thought to be caused by a dysbiotic oral microbiota characterized by increased levels of *Streptococcus* and *Lactobacillus* in the oral cavity [64]. Another study described a different oral microbiota in HIV-infected subjects, characterized by a lower microbial diversity, and enrichment in *Veillonella*, *Rothia*, and *Streptococcus* spp. [65].

### 2.3. Respiratory Tract Microbiota

In the past, it was the assumption that lungs were sterile; however, huge advances in culturing techniques demonstrated that the microbial colonization of the respiratory tract begins in utero. After birth, the respiratory tract is colonized with the mother’s microbiota. In a similar manner to the skin microbiota, the diversity of the respiratory microbiota highly depends on the mode of delivery. Natural delivery prompts the colonization of microbes from the mother’s vaginal and gut microbiota, while C-section newborns are colonized by the mother’s skin microbiota [66,67]. Studies on the respiratory microbiota have highlighted the limitations in the determination of a core respiratory microbiota, due to the great interpersonal variability. Nevertheless, data indicates that certain bacteria general such as *Streptococcus*, *Haemophilus*, *Moraxella*, *Staphylococcus*, and *Veillonella* are commonly presented in samples of the respiratory microbiota [68].

The respiratory tract can be divided into two parts, the upper and the lower respiratory tract. They both are attached to each other, however, they present different environmental conditions (pH, temperature, PCO_2_, and PO_2_ conditions) [69], and also harbor different bacterial communities [70]. The upper respiratory tract, which can be divided into nasal cavity, nasopharynx, and oropharynx, contains most of the bacteria, that are predominatly *Staphylococcus*, *Propionibacterium*, *Corynebacterium*, *Streptococuus*, *Moraxella*, *Haemophillus*, *Prevotella*, and *Veillonella* [69]. The lower respiratory tract includes the trachea and lung’s bronchial trees, and is mostly represented by *Prevotella*, *Veillonella*, *Streptococcus*, and *Tropheryma* [69]. The bacteria density decreases as we descend in the tract, being the lungs the location with the lowest bacterial count [68,69].

### 2.4. Gut Microbiota

The gut microbiota is by far the most studied of the microbiota niches in the human body, and this is because it contains around 70% of the human microbiota [71]. The microbiota in this area is not evenly divided, and the microbial composition and relative abundance change according to the section of the digestive tract [10]. This can partially be explained by the chemical, nutritional, and immunological gradient along the digestive tract [71]. As with other microbiota communities, the gut has first contact with microorganisms after delivery, and is deeply influenced by environmental determinants such as early life events [19] such as the delivery mode [15], or breastfeeding [17]. Normally, it is first colonized by facultative anaerobes, and there is a gradual shift towards anaerobes species [72].

Although the microbiota is present all along the gastrointestinal tract, the greatest number of bacteria is concentrated in the large intestine, specifically in the colon [73], where bacteria of the phylum Bacteroidetes and Firmicutes predominate, representing 90% of the gut microbiota [10]. Members of the *Bifidobacterium*, *Lactobacillus*, *Bacteroides*, *Clostridium*, *Escherichia*, *Streptococcus*, and *Ruminococcus* genera are among the most representative intestinal microbes [74].

There is a long list of factors that influence the gut microbiota composition. Some notable examples are the host genome [12,30], geography [75], adulthood diet [20], physical activity [27], host’s circadian clock [28], and psychological stress [26]. Nevertheless, the dietary factors are probably the strongest and more powerful determinants shaping the gut microbiota. The characteristics of the diet have a profound effect on the gut microbiota profile, affecting both composition and diversity [72,76]. The dietary pattern, the contribution of the macronutrients, the presence of bioactive components or functional food, or the use of nutraceuticals such as probiotics and prebiotics can effectively alter the microbiota composition and confer health benefits to the host [74,77].

The relationship between gut and brain has been extensively studied as well. This gut–brain axis is very important due to the role of gut’s microbiota has on behavior and development of the brain [78]. However, the absence of microbiota in the brain means that the hypothesis presented in this review does not fit on this axis, since the relationship between the brain and the intestine occurs through metabolites that are capable of crossing the blood–brain barrier. This evidence has been highlighted in different studies, in which a relationship was found between an altered intestinal microbiota and an affected brain [79]. It is crucial to mention the relationship established between the main neurodegenerative disorders, Parkinson disease (PD) and Alzheimer’s disease (AD), and a gut’s microbiota dysbiosis. In the first example, it seems that an overgrowth of *Helicobacter pylori* on the GI tract is linked to a severe form of the PD. In addition, an increase of pro-inflammatory bacteria is linked to PD, these bacteria are *Proteobacteria*, *Enterococcus*, and *Enterobacteriaceae*. Similar results have been reported in AD, where a decrease of *Eubacteria* (*E. rectale*), which is anti-inflammatory, and an overgrowth of *Escherichia* and *Shigella*, pro-inflammatory, lead to an aggravation of the disorder [80]. The communication between both systems can be divided into five pathways, neuroanatomical pathway, neuroendocrine mediated by hypothalamic–pituitary–adrenal axis, gut immune system, neural regulators synthesized by gut bacteria and intestinal and blood–brain barrier [78].

### 2.5. Genital Microbiota

The vaginal microbiota is simpler than other microbiota niches, for instance, the gut microbiota, and presents lower alpha and beta diversity [81]. It is governed by *Lactobacillus* spp., mostly *L. crispatus*, *L. iners*, *L. gasseri*, and *L. jensenii*, which exert an important defensive function, and other species from the genera *Atopobium*, *Dialister*, *Gardnerella*, *Megasphaera*, *Prevotella*, *Peptoniphilus*, *Veinovella*, *Lachnospiraceae*, *Streptococcus*, *Staphylococcus*, and *Gemella*, among others [82,83]. The stability of the human female microbiota is known to fluctuate during lifespan. Indeed, due to the great endogenous and exogenous fluctuations during the menstrual cycle, the vaginal microbiota, and particularly lactobacilli, also fluctuate during the period [84]. During menopause, however, the drop in estrogen levels has been associated with a decline in *Lactobacillus* spp. and genitourinary complications such as urinary tract infections [85]. In addition, during pregnancy the hormonal changes and the many physiological and structural alterations influence the vaginal microbiota. According to cross-sectional studies, it exhibits important changes including a decline in alpha diversity, increased number of *Lactobacillus* spp., particularly *L. iners*, *L. crispatus*, *L. jensenii*, and *L. johnsonii*, increased abundance of Clostridiales, Bacteriodales, and Actinomycetales, as well as changes in the profile of microbial metabolites produced by the vagina microbiota [86,87]. Interestingly, microbiota variations during pregnancy are not restricted to the vaginal tract since compositional changes have also been reported in the oral and gut microbiota of pregnant women [81,86,88]. To illustrate, it has particular relevance the greatest rate of the periodontal disease reported in pregnant women that is itself linked to preterm birth [81,88].

The vaginal microbiota, and particularly *Lactobacilli* spp., plays a vital role in the female reproductive fitness and pregnancy outcome. To illustrate this, *Gardnerella vaginalis* and *Atopobium vaginae* have been associated with a poor pregnancy rate [88]. The studies on the vaginal microbiota vastly outnumbering the number of studies on the penis microbiota, and many of them are related to bacterial vaginosis (BV). This is the most common genital tract infection in women and is characterized by greater bacterial diversity, an enrichment in anaerobic and facultative bacteria species from the genera *Atopobium*, *Gardnerella*, *Mycoplasma*, *Prevotella*, *Bifidobacterium*, *Megasphaera*, *Leptotrichia*, *Sneathia*, *Dialister*, or *Clostridium*, as well as a reduced number of *Lactobacilli* normally found in healthy women [83,89,90,91,92]. The BV has been extensively studied, and findings from a large number of human studies concluded that, in the majority of the studied population, a set of women had a microbiota enriched with *L*. *iners* or *L*. *crispatus*, and that women in the second group presented protection against developing a vaginal microbiota prone to BV than those in the first group [83]. Two particular bacteria, *G. vaginalis* and *A. vaginae*, have gained increasing attention and are among the main bacteria involved in BV. Their pathogenicity seems to be related to their ability to establish microbial biofilms with other species [89,91,93]. Certain probiotic strains (*L. reuteri* RC-14 and *L. rhamnosus* GR-1) have been demonstrated to impair those biofilms and showed promise as potential therapeutic agents for the restoration of the normal vaginal microbiota in women with BV [89].

In the same line, vaginal dysbiosis can also be associated with vulvovaginal candidiasis, which is the overgrowth of *Candida* spp. *Candida albicans* is the most frequent species; however, other *Candida* spps. like *C. tropicalis*, *C. glabrata*, *C. krusei*, *C. dubliniensis*, and *C. parapsilosis* have been identified [94]. *C. albicans* is a commensal fungi that is naturally present in the oral, gut, and vaginal microbiota; however, when there are imbalances in the microbiota composition and there is a drop in certain bacteria groups, *Candida* spps. take advantage, expand, and behave has a pathobiont causing oral, vaginal, or intestinal inflammation and candidiasis [95]. The risk factors for developing vaginal candidiasis are many, including hormonal environment, personal hygiene, exposition to antibiotics or antifungal agents [94]. On the other hand, the presence of certain *Lactobacillus* spp., such as *L. crispatus*, has been negatively associated to vaginal candidiasis and showed a protective for BV and STIs too [94].

A dysbiotic vaginal microbiota has been associated with infections by human papillomavirus or human immunodeficiency virus, risk of suffering BV and STIs, infertility, and also female reproductive health complications such as septic postpartum, neonatal infections, or miscarriage [83,87,88]. Moreover, the vaginal microbiome affects the success rates of in vitro fertilization, and the characteristic of the microbial communities in the placenta and the amniotic liquid importantly affect the pregnancy and reproductive outcome [88]. Therefore, research effort should concentrate to improve understanding of the conditioning factors of the female microbiota and the consequences of its perturbations.

Published studies on the penis microbiota are still relatively limited in number; however, it has potential health implications. The composition of the penile microbiota skin is dramatically affected by circumcision, including a decrease in anaerobic bacteria counts, and such changes seem to have a protective effect against STIs like human papillomavirus and human immunodeficiency virus [96]. One study in Black South African observed that the penis microbiota was dominated by *Corynebacteriaceae*, *Prevotellaceae*, *Clostridiales*, *Porphyromonadaceae*, and *Staphylococcaceae* families. Most subjects presented a microbiota enriched in *Corynebacterium* spp. [96]. Results from another study indicate that those men with a high presence of anaerobic bacteria in the penis have a greater risk for acquiring HIV as compared to those men with a healthy microbiota [97]. On top of that, data suggests that men and women share genital microbiota during heterosexual intercourse [88], and thus the penile microbiota may also have a key effect on women’s urogenital impact. Moreover, an interesting study indicated that the profile of the penile microbiota could predict the risk for BV in women [49].

### 2.6. Urinary Microbiota

In contrast to other microbiota reservoirs, the female urine microbiota has been poorly investigated. Hopefully, in the last years, there has been a rapid rise in interest in describing its composition. According to one clinical trial, the healthy female urinary microbiota can be categorized into urotypes according to the relative abundance of *Lactobacillus*, *Gardnerella*, *Sneathia*, *Staphylococcus*, and *Enterobacteriacae* members [98]. Frequently, *Lactobacillus* is the dominant genus [48]. In an American multi-ethnic population of women aged 35–75 years, the urinary microbiome of women with urgency urinary incontinence differed from that in controls. The results indicate that the microbiota analyzed was poor in *Lactobacillus* spp., which are of great importance for the bladder health [99], and enriched in members of the genera *Actinobaculum*, *Actinomyces*, *Aerococcus*, *Arthrobacter*, *Corynebacterium*, *Gardnerella*, *Oligella*, *Staphylococcus*, and *Streptococcus*, and certain bacteria species (*Actinobaculum schaalii*, *Actinomyces neuii*, *Aerococcus urinae*, *Arthrobacter cum- minsii*, *Corynebacterium coyleae*, *Gardnerella vaginalis*, *Oligella urethralis*, and *Streptococcus anginosus*), some of which are uropathogens [98]. Besides, both groups presented a distinct profile of Lactobacillus spp., being that *L. gassesi* was more characteristic in cases, while *L. crispatus* was more represented in controls [98].

As in the case of the female urinary microbiota, the male urinary microbiota has been hardly investigated. Besides there was a great intra-subject variability, one study in a sample of sexually active men indicated that the male urinary microbiome (urobiome) was mostly represented by Firmicutes, followed by other phyla such as Actinobacteria, Fusobacteria, Proteobacteria, and Bacteroidetes, and underrepresented by Tenericutes and TM7 [100]. Interestingly, the analysis revealed that the majority of the identified microbial groups matched to species from the female urogenital tract, and that, to some extent, the composition resembles that in other body regions such as the skin or the colon [100]. The analysis also indicated that sexually transmitted infections (STIs) by pathogens such as *C. trachomatis* and *N. gonorrhoeae*, are associated with a urine microbiota poor in terms of genera diversity. It was mostly represented by *Lactobacillus*, *Corynebacterium*, *Streptoccus* and *Sneathia* spp., and other taxa like *Aerococcus*, *Anaerococcus*, *Prevotella*, *Gemella*, *Veillonella*, and *Sneathia* spp. were less representative. This dysbiotic microbiota was also linked to a greater risk for STI or bacterial vaginosis in women [100]. These authors have also suggested that urine samples offer a good representation of the male urinary microbial community, particularly the urethral epithelium, and therefore show promise for the diagnosis of sexually transmitted infections (STI) in male subjects [101]. Though current knowledge is limited, it may be possible to screen the risk for STIs using microbiota samples in a near future. This is important since it has been established an association between the dysbiotic microbiota in STIs and BV and HIV [101,102]. Besides that, the urinary microbiota is suspected to relate to prostate cancer. A detailed study showed a greater prevalence of pro-inflammatory bacteria and uropathogens in urinary samples from men with prostate cancer [103].

As with women, men’s reproduction could be also influenced by the microbiome. One pilot study reported that seminal microbiota from infertile men has a greater α-diversity and differs from rectal samples in terms of β-diversity, is enriched in *Aerococcus* and poor in *Collinsella* [104]. Besides, some bacteria genera were linked to features of sperm quality such as sperm concentration or total motile sperm count [104]. On top of that, men’s infertility seems to influence other microbiota niches, and the rectum microbiota in this population had a drop in Anaerococcus and enrichment in *Lachnospiraceae*, *Collinsella*, and *Coprococcus*, while the urinary microbiota was rich in *Anaerococcus* members [104].

The main information of this section has been summarized in Table 1.

## 3. The Interplay between the Different Microbiotas

Just as there is communication between the human cells, bacteria also communicate between the different niches where they are established [118], as well as with the human cells [119,120]. As expected, since its harbors the vast majority of the microorganisms, the gut microbiota is the main core of communication, and it seems plausible that the main interplay is established between gut microbiota and the others through the well-described gut-liver axis [121], gut–brain axis [122], gut–skin axis [123], and oral–gut axis [124]. On the other hand, other authors have shown crosstalk between different regions in which the gut microbiota is not involved, such as the oral–pulmonary axis [69].

When speaking about microbiota communication, secondary metabolites deserve special attention. They represent a way of communication between bacteria but also have a key role in the regulation of the host’s immune system [114,125]. Metabolites may distribute to distant sites of the organism by entering the circulation [113,126], traveling throughout the blood, and finally could accumulate in other regions, perturbing the health of the target zone. The short-chain fatty acids (SCFAs), such as butyrate, propionate, acetate, or lactate are the best studied and the most prominent immunomodulatory metabolites [127]. They can exert pleiotropic effects on several body sites, influencing the normal functionality of the liver, gut, or pancreas [127]. SCFAs are a by-product of fiber fermentation by certain intestinal microorganisms, being *Roseburia intestinalis*, *Faecalibacterium prausnitzii*, *Eubacterium hallii*, *Bacteroides uniforms*, *Prevotella copri*, *Akkermansia muciniphila*, *Bifidobacterium* spp., and *Lactobacillus* spp. the most important SCFAs producers [113,126]. There are other metabolites produced by the gut microbes that are considered biomarkers of a disturbed gut, such as free phenol and p-cresol [126,128]. As well as metabolites, previous evidence suggests that bacteria themselves also could enter the circulation due to a disturbed intestinal barrier function, causing, once again, damage into the zone where they move [128]. As was mentioned above, the different microbiotas which inhabit the human body may create various axes forming a net with a cross-talk between all of them, mainly through microbial-derived metabolites. Nevertheless, it is a hypothesis, and more evidence is needed to clarify the underlying mechanisms of microbiota communication [61].

Previous studies have demonstrated the importance of the gut microbiome over skin health, and some of them suggest that imbalances in gut–skin axis could lead to inflammatory skin diseases like atopic dermatitis (AD) [126]. In addition, it was reported a link between low intestinal microbial diversity and AD, that was attributed to a reduced and abnormal immune maturation in childhood [105]. In the same way, the reduction of the gut microbiota diversity is also present in skin disease as psoriasis [106].

Another axis that has been studied extensively is the oral–gut axis, and available evidence suggests that the oral microbiota has a great influence on the intestinal one. A possible explanation is that the oral microbiota can affect the gut by the dissemination of some bacteria, such as *Poryphyromonas*, *Fusobacterium*, *Oscillibacter*, *Peptostreptococcus*, *Roseburia*, and *Ruminococcus*, which are periodontal pathogens and have also been found in samples from patients with colorectal cancer (CRC) [61]. Other members of the oral microbiota (*Veillonella* and *Streptococcus*) are thought to be involved in the development of liver cirrhosis, a disease related to intestinal dysbiosis, demonstrating that oral microbiota may also affect intestinal microbiota and ultimately the liver [108].

Further evidence supporting the link between different microbiotas is the case of rheumatoid arthritis, where both oral and gut microbiota are disturbed and seem to be contributing factors in the disease development [58]. The similar has been described for the SARS-CoV-2. A recent study concluded that the virus could promote an oral dysbiosis, probably because the oral cavity serves as SARS-CoV-2 reservoir [129]. Previous reports have indicated that the infection also promotes a proinflammatory status in the lungs and has an impact on the lung microbiota, that present greater levels of *Klesiella oxytoca*, *Faecalibacterium prausnitzii*, and *Rothia mucilaginosa* [110]. This oral dysbiosis could lead to the translocation from the oral cavity to the digestive tract, resulting in gut inflammation and dysbiosis, both of with are frequently observed in subjects presenting COVID-19 [46,130]. COVID-19′s severity was associated to a characteristic gut microbiota profile with higher levels of *Coprobacillus*, *Clostridium ramosum*, *and Clostridium hatheway* and lower number of *Faecalibacterium prausnitzii* [110,111]. Moreover, the composition of the gut microbiota changed during the progression of the disease, and some of these alterations remain after the resolution of infection [131]. It should be noted that bacterial co-infection occurred in 7% of hospitalized COVID-19 patients. Compared with patients in mixed wards/intensive care unit (ICU) settings, ICU COVID-19 patients have a higher proportion of bacterial infections [132]. These studies suggested that high vigilance should be stablished against infections derived from the oral microbiome during infection by respiratory viruses such as SARS-CoV-2. Uncovered risk factors such as increased inhalation, poor oral hygiene, and viral infection have been related to the occurrence of respiratory infection [133,134]. The mechanisms by which the oral microbiome can influence respiratory disease such as COVID-19 is complicated and multifactorial, simultaneously affected by environmental, host, and microbial factors [135,136].

There are other examples that support the hypothesis of the gut–lung axis, since asthma, chronic obstructive pulmonary disease (COPD), cystic fibrosis, and lung cancer have been associated with important alterations in the gut microbiota composition [109,112,113,114]. Interestingly, the oral dysbiosis may also affect distant sites and produce systemic complications, such as the case of systemic lupus erythematosus, where the diversity of oral microbiota is compromised, and membres of *Lactobacillaceae*, *Veillonellaceae*, and *Moraxellaceae* families are increased [107].

Previous studies have suggested a cross-talk between gut and lung microbiota, and that a previous gut disturbance may be responsible for subsequent lung diseases [113,114]. As in the case of another microbiota axis, the gut microbiota has a relevant role in this communication, while the contribution of the lung microbiota remains to be elucidated [137]. For instance, it has been demonstrated that gut dysbiosis is accompanied by the secretion of SCFAs from the gut microbiota to the lungs, which causes lung inflammation and a major susceptibility to allergens [114].

Two body locations that also are in close contact are the oral cavity and the lung. Both the oral and the lung microbiota present some similarities that could be explained by their communication through the respiratory tract. Indeed, considering that, it seems plausible that the lung microbiota has origin in the oral one [138]. For example, it has been associated with poor oral health could contribute to asthma or pneumonia [129,139], and changes in the oral microbiota were reported in HIV-infected subjects [65]. Previous studies have suggested a relationship between oral dysbiosis and lung disease; however, the mechanisms involved are still not fully understood. Further findings suggest that oral bacteria may communicate the lung through by inflammatory proteins; however, this issue remains unclear [124].

Previous studies also support the idea that there exists a gut–lung axis that allows for the exchange of molecules (microbial metabolites, hormones, toxins, proteins) between the gut and the lung, mainly through the systemic circulation [114]. Compelling evidence suggest that such interactions, that importantly influences the immune and inflammatory states, are implicated in different lung diseases including infections (tuberculosis, pneumonia), genetic diseases (cystic fibrosis), inflammatory diseases (asthma, COPD), and cancer (lung cancer) [113,114]. Even though each disease was associated to different disturbances in the intestinal microbiota (for details see [114]), it was observed an overgrowth of Proteobacteria and Firmicutes taxa in these cases. To illustrate, several studies have linked gut dysbiosis in early life to asthma, condition in which the genus *Faecalibacterium* and *Roseburia* are present in lower proportion, and other bacteria genera are enriched as compared to healthy individuals [109]. In addition, gut microbiota has been associated with the development of lung cancer, with some studies indicating that the use of antibiotics before and during the therapy can decrease the efficiency of the antitumor drugs due to the interaction between antibiotics and gut microbiota, that is strongly affected by xenobiotics [112]. Although the causality remains to be clarified, the available information strongly suggests that the gut microbes play a critical role in lung health, and therefore should be contemplated in lung disease’s prevention and treatment.

It has also been investigated the communication between the gut and the liver by means of the gut–liver axis. As example, SIBO was found in more than half of subjects with liver cirrhosis, and was associated to systemic endotoxemia [28]. Besides, another group identified a different microbiota profile in controls and patients with liver cirrhosis, which was characterized by members from oral origin [108]. Indeed, these authors developed a discrimination index with gene markers from the intestinal microbiota.

Due to their proximity within the human body, it is plausible that the urinary microbiota affects the genital one. One study hypothesized that there may be a urogenital microbiome that comprises microorganisms from both the urinary and the vaginal tracts [98], and another study also confirmed the presence of both urinary and genital microorganisms in the urine [116]. A further cross-sectional study on more than 200 women corroborated that urinary and vaginal microbiome share more than half of the most abundant operational taxonomic units. Both microbiota niches were dominated by *Lactobacillus*, especially the vaginal niche, and presented varying levels of *Gardenerella*, *Prevotella*, and *Ureaplasma* [117]. In the same line, it has been reported that the urinary microbiota from women suffering BV clustered differently to that of healthy women, and differences persisted following the antibiotic treatment [116]. A previous study pointed that bacteria genera frequently identified in episodes of BV are also naturally found in the bladder of healthy women, suggesting the transference of microorganisms from the urinary to the genital tract [83].

Interestingly, other microbiota niches could be involved in the development of BV and dysbiotic vaginal microbiota. A relevant publication reporting data on a prospective cohort study of young women who reported sex with other women suggested that women presenting certain bacteria in the oral cavity (*G. vaginalis*) or anal samples (*G. vaginalis* and *Leptotrichia/Sneathia* spp.) are more likely to suffer from BV [90]. In the same line, a recent report on young South African females with a high prevalence of BV indicated that dysbiosis in the oral and vaginal microbiota are frequently concurrent and that the oral cavity of women presenting vaginal dysbiosis was enriched in bacteria members linked to periodontal disease [92]. Another relevant study corroborated the correspondence between the oral, vaginal, and rectum microbiota [140]. The information above offers novel potential targets to restore vaginal dysbiosis and therefore decrease the risk for adverse life events previously mentioned.

Human microbiome analysis has been largely based on observation, with associations of disease phenotypes with particular microbiota constituents. However, one of the most controversial points in the study of the human microbiota is to establish whether the presence of a certain population of microorganisms is a cause or effect of the underlying disease and how this change can affect other niches where a specific microbiota resides. Different mechanisms can explain this connection, from metabolites (such as SCFA) to part of bacteria (such as extracellular bacterial vesicles) that migrate from different parts of the human body, to even the bacteria themselves that can cross epithelial barriers (such as the intestine epithelial cells) that lose their integrity in disease conditions (such as obesity). Extracellular bacterial vesicles have caught the attention of researchers [121,122] as one of the mechanisms by which distant microorganisms could communicate, as it has been shown to occur with exosomes as an intercellular communication system in multicellular organisms.

Changes in the local microbiota occur in close contact with nearby cells, both host cells (with which there is a symbiotic or commensal relationship) and with nearby microorganisms with which they compete for the location and the nutrients in their environment. In this sense, the equilibrium that occurs is dynamic depending on multiple factors, both intrinsic (metabolism of the microorganisms present) and extrinsic (nutrients, pH conditions, oxygen pressure) that ultimately modulate the local microbiota present in a certain organ. In turn, the host cells are also influenced by the presence of a certain microbiota and respond to it by adapting in a truly dynamic equilibrium that, when disrupted, is responsible for the development of a disease.

The main information of this section has been summarized in Figure 1.

## 4. Conclusions

In conclusion, although there is a lack of evidence in the field of microbiota communication, several studies have emphasized the influence of the gut microbiota on microbiota located in other parts of the body. Regarding this aspect, there are theoretical grounds for believing that the gut microbiota plays a more active role in the host phenotype. As with any cell in a biological system, it could be that the microbiota is a well-organized and structured network in which the intestinal microbiota behaves as a central regulator that integrates peripheral microbiota. Because of the foregoing, the different microbiotas become potential approaches to investigate, so that restoring a particular microbiota system may indirectly lead to improvements in a distant microbiota and thus confer health improvements to the host. This new approach would provide new therapeutic strategies.

Nevertheless, we are aware that to date, the intestinal microbiota has been one of the most widely researched, and at present, there is insufficient research on other microbiota regions to prove this hypothesis. In addition, mechanistic studies are lacking and the resources required for these experiments have not been well established. This is a compelling area for future research, and to achieve this objective microbiota research should focus on a much more integrative model that takes into account the target microbiota but also other supposedly unrelated microbiotas. For that purpose, multi-omics approaches and appropriate bioinformatics analysis appear indispensable.

To the best of our knowledge, this is the first report to hypothesize the potential interplay and crosstalk between the different human microbiotas. We would be pleased if our contribution opened a new door to a better understanding of the relationship between host health and microbiota.

## Figures and Tables

**Figure 1 nutrients-13-02905-f001:**
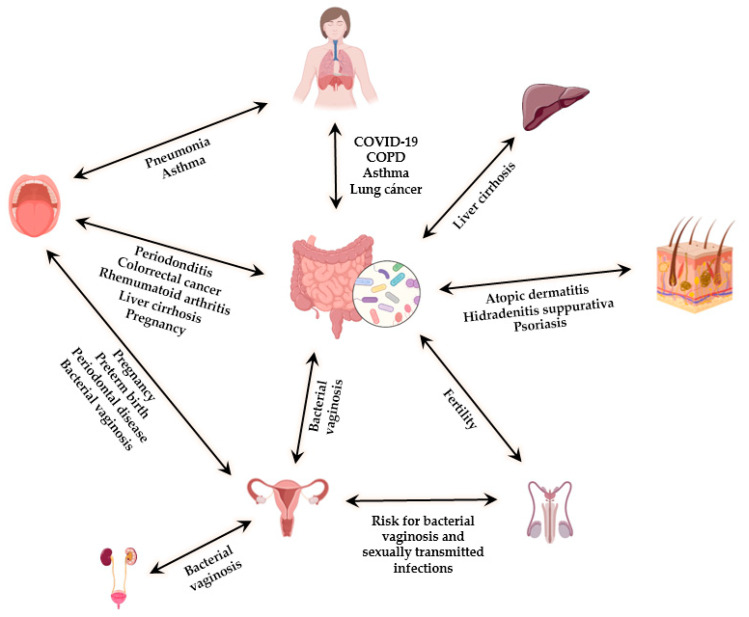
Schematic illustration representing the main microbiota niches in the human body, possible interaction between them, and related diseases.

**Table 1 nutrients-13-02905-t001:** Summary of the main microbiota niches in the human body and potential connection with other microbiotas and body sites.

Microbiota Niche	Predominant Taxonomic Groups	Associated-Diseases or Conditions and Characteristic Microbiota Composition (If Available)	Potential Communication with Other Microbiota Niches/Organs
**Skin microbiota**	Actinobacteria, Firmicutes, and Proteobacteria phyla [50]. *Corynebacterium*, *Propionibacterium*, *Staphylococcus*, and *Streptococcus* genera [50,51].	Natural delivery: *Lactobacillus* and *Prevotella* spp. [15]. C-section delivery: *Propiniobacterium*, *Corynebacterium*, and *Staphylococcus.* Acne: ↑ *Propionibacterium* spp. (*Propionibacterium acnes*) [50,56]. Atopic dermatitis: *S. aureus* spp. (*S. aureus* and *S. epidermidis*) and *Malassezia* spp. fungi frequently found in cases [51,57], Hidradenitis suppurativa: ↑ *Corynebacterium*, *Porphyromonas*, and *Peptoniphillus* spp. [59]. Seborrheic dermatitis: *Malassezia* spp. [50]. Rosacea: *Demodex* mite [50].	Gut–skin: ↓ intestinal microbial diversity in atopoic dermatitis [105] and psoriasis [106].
**Oral microbiota**	Actinobacteria, Bacteroidetes, Firmicutes, Proteobacterias, and Synergistetes phyla [62] *Fusobacterium*, *Gemmela*, *Veillonella*, *Streptococcus* and *Granulicatella* genera [63] Saliva: *Streptococcus*, *Veillonella*, and *Prevotella* spp. [60] Soft tissues: *Streptococcus salivarus*, *Rothia*, and *Eubacterium* spp. [60]. Tooth: *Corynebacterium*, *Actinomyces*, *Spirochaetes*, *Fusobacteria*, *Actinobacteria*, *Proteobacteria*, and *Bacteroidetes* spp. [60]	Caries: ↑ *Streptococcus* and *Lactobacillus* spp., associated with oral candidiasis [64]. Rheumatoid arthritis: ↑ diversity and differential microbiota composition (for details see [58]). Osteoartritis: ↑ diversity and differential microbiota composition (for details see [58]). Systemic lupus erythematosus: ↓ diversity, ↑ *Lactobacillaceae*, *Veillonellaceae*, and *Moraxellaceae* families [107]. HIV: ↓ diversity, ↑ *Veillonella*, *Rothia*, and *Streptococcus* spp. [65]. BV: ↑ bacteria members associated with periodontal disease [92].	Oral cavity–gut–vagina: compositional changes in the oral and gut microbiota of pregnant women [81,86,88]. Oral cavity–vaginal: ↑ Rates of periodontal disease in pregnant women, associated to preterm birth [81,88]; identification of *G. vaginalis* in the oral cavity was associated with a greater risk for BV [90]. Oral cavity–gut: presence of periodontal pathogens (*Poryphyromonas*, *Fusobacterium*, *Oscillibacter*, *Peptostreptococcus*, *Roseburia*, and *Ruminococcus* spp.) in intestinal samples from colorectal cancer [61]; presence of members of the oral microbiota in gut samples of patients with liver cirrhosis [108].
**Respiratory tract microbiota**	Bacteroidetes, Actinobacteria, and Firmicutes [109] *Streptococcus*, *Haemophilus*, *Moraxella*, *Staphylococcus*, and *Veillonella* spp. [68]. Upper respiratory tract (nasal cavity, nasopharynx, and oropharynx): *Staphylococcus*, *Propionibacterium*, *Corynebacterium*, *Streptococuus*, *Moraxella*, *Haemophillus*, *Prevotella*, and *Veillonella* spp. [68]. Lower respiratory tract (trachea and lung’s bronchial trees): *Prevotella*, *Veillonella*, *Streptococcus*, and *Tropheryma* spp. [68].	COVID-19: ↑ *Klebsiella oxytoca*, *Faecalibacterium prausnitzii* and *Rothia mucilaginosa* [110].	Gut–lung: COVID-19 associated with ↑ *Coprobacillus*, *Clostridium ramnosum*, and *Clostridium hathewayi*, ↓ *Faecalibacterium prausnitzii* in faecal samples [110,111]; asthmatic presented different compositional characteristic in the gut microbiota [109,112].
**Gut microbiota**	Bacteroidetes and Firmicutes phyla [10]. *Bifidobacterium*, *Lactobacillus*, *Bacteroides*, *Clostridium*, *Escherichia*, *Streptococcus* and *Ruminococcus* spp. [74].	Alzheimer’s disease: ↓ *E. rectale*, ↑ *Escherichia* [113,114]. Asthma: ↑ *Bacteroides fragilis*, ↓ *Escherichia coli*, *faecalibacterium*, *Lachnispira*, *Rothia*, *Veillonella*, *Akkermansia municiphila* [113,114]. COPD: ↑ *Enterobacter cloacae*, *Citrobacter*, *Eggerthella*, *Pseudomonas*, *Anaerococcus*, *Proteus*, *Clostridium difficile* and *Salmonella* [113,114] Cystic fibrosis progression: ↑ *Ruminococcus gnavus*, *Enterobacteriaceae*, ↓ *Faecalibacterium prausnitizii*, *Bifidobacterium adolescentis*, *Eubacterium recatale*, *Streptococcus*, *Staphylococcus*, *Veillonela dispar*, *clostridium difficile*, *Pseudomonas aeruginosa*, *Escherichia coli* [114]. Liver cirrhosis: different intestinal composition vs. controls [108]. Lung cancer: ↑ *Enterococcus*, ↓ *Actinobacteria* and *Bifidobacterium* [112]. Parkinson’s disease: *Helycobacter pylori infections* and ↑ *Proteobacteriam Enterococcus* and *Enterobacteriacea* [80]. Rheumatoid arthritis: ↑ diversity and differential microbiota composition (for details see [58]). Osteoartritis: ↑ diversity and differential microbiota composition (for details see [58]). Pulmonary diseases: ↑ Proteobacteria and Firmicutes [114].	Gut–liver: SIBO was found in more than half of subjects with liver cirrhosis, and was associated to systemic endotoxemia [115]. Gut–vagina: identification of *Gardnerella vaginalis* and *Leptotrichia/Sneathia* spp. in rectal microbiota samples was associated with greater risk for BV [90].
**Vaginal microbiota**	*Lactobacillus* spp., (*L. crispatus*, *L. iners*, *L. gasseri*, and *L. jensenii*), *Atopobium*, *Dialister*, *Gardnerella*, *Megasphaera*, *Prevotella*, *Peptoniphilus*, *Veinovella*, *Lachnospiraceae*, *Streptococcus*, *Staphylococcus* and *Gemella* [82,83].	BV: ↑ bacterial diversity, ↑ anaerobic and facultative bacteria species from *Atopobium* (*A. vaginae*), *Gardnerella* (*G. vaginalis*), *Mycoplasma*, *Prevotella*, *Bifidobacterium*, *Megasphaera*, *Leptotrichia*, *Sneathia*, *Dialister*, *Clostridium* spp.), *Lactobacilli* spp. [83,89,90,91,92,93]. Placenta and the amniotic liquid microbiota influences Pregnancy: ↓ α-diversity, ↑ *Lactobacillus* spp. (*L. iners*, *L. crispatus*, *L. jensenii*, and *L. johnsonii*), *Clostridiales*, *Bacteriodales*, and *Actinomycetales*, differential profile of microbial metabolites [86,87] Pregnancy and reproductive outcome [88]. Poor pregnancy rates: *Gardnerella vaginalis* and *Atopobium vaginae* [88]. *Lactobacillus crispatus* gives protections against BV and STIs [83] Lactobacillus *iners* is associated to a greater risk for bacterial vaginosis [83]. Vaginal dysbiosis associated to septic postpartum, miscarriage or neonatal infections [83,91]. Vaginal dysbiosis associated to infertility [88].	Vagina–oral cavity: compositional changes in the oral and gut microbiota of pregnant women [81,86,88]; oral dysbiosis in women with BV [92]. Vagina–bladder: presence of urinary and genital microorganism in urinary samples [116]; similar composition in urinary and vaginal microbiota samples [117].
**Penile microbiota**	*Corynebacteriaceae*, *Prevotellaceae*, *Clostridiales*, *Porphyromonadaceae*, and *Staphylococcaceae* families [96]. *Corynebacterium* spp. [96]	Circumcision: ↓ anaerobic bacteria and protective effect against STIs (HPV and HIV) [96,97]. Greater risk for HIV: ↑ anaerobic bacteria [97]. Infertility: ↑ α-diversity, ↑ *Aerococcus* spp., ↓ *Collinsella* spp. [104]. Some bacteria correlates to sperm quality [104].	Penis–vagina: penile microbiota (*Parvimonas*, *Lactobacillus iners*, *Fastidiosipila*, *Negativicoccus*, *L. crispatus*, *Dialister*, *Sneathia sanguinegens*, *Gardnerella vaginalis*, *Prevotella corporis*, and *Corynebacterium*) can predict the risk for BV in women [49]. Semen–urine–rectum: ↓ *Anaerococcus*, ↑ *Lachnospiraceae*, *Collinsella*, and *Coprococcus* spp. in rectum microbiota in infertile men [104].
**Female urinary microbiota**	*Lactobacillus* spp. [48].	Urgency urinary incontinence: ↓ *Lactobacillus* spp. (*L. crispatus*) [98,99], ↑ *Actinobaculum*, *Actinomyces*, *Aerococcus*, *Arthrobacter*, *Corynebacterium*, *Gardnerella*, *Oligella*, *Staphylococcus*, and *Streptococcus genera*, ↑ *Actinobaculum schaalii*, *Actinomyces neuii*, *Aerococcus urinae*, *Arthrobacter cumminsii*, *Corynebacterium coyleae*, *Gardnerella vaginalis*, *Oligella urethralis*, and *Streptococcus anginosus* [98].	Bladder–vagina: different urinary microbiota in women with BV [116]; presence of bacteria genera natural from the bladder in ephisodes of BV [83].
**Male urinary microbiota**	Firmicutes, Actinobacteria, Fusobacteria, Proteobacteria, and Bacteroidetes [100]. *Lactobacillus* (*L. iners*), *Aerococcus*, *Anaerococcus*, *Prevotella*, *Gemella*, *Veillonella*, *Sneathia*, *Corynebacterium*, *Staphylococcus* and *Streptococcus* spp. [100,103].	STIs (*Chlamydia trachomatis* and *Neisseria gonorrhoeae*); ↓ diversity at genus level. Prostate cancer: ↑ pro-inflammatory bacteria and uropathogens [103]. *Gardnerella vaginalis* was associated with chronic inflammation in prostate biopsies [103].	Urine(bladder)–skin–colon–vagina: the composition of male urine resembles that in the skin, colon, and vagina [100]. Urine–vagina: male urinary microbiota of subjects infected with STIs is linked to a greater risk for STI or bacterial vaginosis in women [100].

BV: bacterial vaginosis; COVID-19: coronavirus disease 2019; HIV: human immunodeficiency virus; HPV: human papillomavirus; SIBO: small intestinal bacteria overgrowth; STIs: sexually transmitted infections.

## Data Availability

Not applicable.

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
