# Peer review of "Human Microbiota Network: Unveiling Potential Crosstalk between the Different Microbiota Ecosystems and Their Role in Health and Disease"

_nutrients, 2021, doi:10.3390/nu13092905_

Round 1

Reviewer 1 Report

In the manuscript entitled: „Human microbiota network: unveiling potential crosstalk between the different microbiota ecosystems and their role in health and disease”, the Authors present a current overview of literature on microbial communities in the human body as well as a discussion on the potential interplays between different microbiotas. The manuscript is well written, undoubtedly stimulating and though provoking.

In my opinion, the work would benefit from adding information on the role of extracellular bacterial vesicles in communication between bacteria and between bacterial and host cells. In fact, the citations of two reviews on this topics (no 120, 121) are included in the manuscript, but this interesting topics unfortunately is not discussed.

Minor points

  1. Please use the full name of the bacterial species when using it for the first time. For example, in page 6, lines 266-267 there is: “To illustrate this, G. vaginalis and A. vaginae have been associated with a poor pregnancy rate [88].” The full names can be found below in the line 278. Please, check the other species names.

Moreover, the sentence cited above seems to be out of context.

  1. Please check spelling of bacterial names. For example there should be Klebsiella instead of Klesiella (Table 1 and line 422) or C. hathewayi instead of C. hatheway (Table 1 and line 427).
  2. Please use italics for the names of bacterial taxa (Table 1, Gut microbiota, penile microbiota)
  3. Please check the text to remove some typo and grammar errors. For example: line 290 “take” instead of “Take”; line 397 “axes” instead of “axis”; line 411 “been” instead of “be”; line 472 “these cases” instead of “the cases” and possibly in the other places.
  4. In the literature section, position 121 there is not correct journal name. Should be “International Journal of Molecular Sciences”
  5. In the section “3. The interplay between the different microbiotas” two sentences contain repeated information: lines 397-388 “The short-chain fatty acids (SCFAs), such as butyrate, propionate, acetate, or lactate are the best studied and the most prominent immunomodulatory metabolites [128].” and lines 390-391 “SCFAs are a by-product of fiber fermentation by certain intestinal microorganisms, being butyrate, propionate, acetate, and lactate the most representative ones [113,127]”. Perhaps in the second case would be better to list the bacterial species which produce SCFs?

Author Response

REVIEWER #1

Comment #1: In the manuscript entitled: “Human microbiota network: unveiling potential crosstalk between the different microbiota ecosystems and their role in health and disease”, the Authors present a current overview of literature on microbial communities in the human body as well as a discussion on the potential interplays between different microbiotas. The manuscript is well written, undoubtedly stimulating and though provoking.

We greatly appreciate the comments of the reviewer who has captured the essence of the manuscript. It is not always easy to shape the main idea that the authors have and translate it into a review manuscript. We have invested many hours in reviewing and discussing the information that, as a final result, has led to the present review. This could not have been possible without the teamwork of the authors signing the manuscript.

Comment #2: In my opinion, the work would benefit from adding information on the role of extracellular bacterial vesicles in communication between bacteria and between bacterial and host cells. In fact, the citations of two reviews on this topics (no 120, 121) are included in the manuscript, but this interesting topics unfortunately is not discussed.

Regarding the reviewer's comment, we have include the following paragraph in the Discussion section where we try to describe the mechanisms behind microbial intercommunication (with special mention of bacterial-derived extracellular vesicles) and whether disease-related phenomena may be the more likely mechanisms responsible for the selection of specific microbial species in different organs:

Human microbiome analysis has been largely based on observation, with associations of disease phenotypes with particular microbiota constituents. However, one of the most controversial points in the study of the human microbiota is to establish whether the presence of a certain population of microorganisms is a cause or effect of the underlying disease and how this change can affect other niches where a specific microbiota resides. Different mechanisms can explain this connection, from metabolites (such as SCFA) to part of bacteria (such as extracellular bacterial vesicles) that migrate from different parts of the human body, to even the bacteria themselves that can cross epithelial barriers (such as the intestine epithelial cells) that lose their integrity in disease conditions (such as obesity,...). Extracellular bacterial vesicles have caught the attention of researchers [121, 122] as one of the mechanisms by which distant microorganisms could communicate, as it has been shown to occur with exosomes as an intercellular communication system in multicellular organisms.

Changes in the local microbiota occur in close contact with nearby cells, both host cells (with which there is a symbiotic or commensal relationship) and with nearby microorganisms with which they compete for the location and the nutrients in their environment. In this sense, the equilibrium that occurs is dynamic depending on multiple factors, both intrinsic (metabolism of the microorganisms present) and extrinsic (nutrients, pH conditions, oxygen pressure,...) that ultimately modulate the local microbiota present in a certain organ. In turn, the host cells are also influenced by the presence of a certain micro-biota and respond to it by adapting in a truly dynamic equilibrium that, when disrupted, is responsible for the development of a disease.

Minor points:

Comment #3: Please use the full name of the bacterial species when using it for the first time. For example, in page 6, lines 266-267 there is: “To illustrate this, G. vaginalis and A. vaginae have been associated with a poor pregnancy rate [88].” The full names can be found below in the line 278. Please, check the other species names.

Moreover, the sentence cited above seems to be out of context.

We appreciate the referee’s comment. The full names of the species have been included, as well as the text revised to modify the names of other possible species that we would have cited in abbreviated form, such as “C. trachomatis” (Chlamydia trachomatis) and “N. gonorrhoeae” (Neisseria gonorrhoeae) (Table 1; Male urinary microbiota). All changes have been highlighted for easy track review.

Comment #4: Please check spelling of bacterial names. For example there should be Klebsiella instead of Klesiella (Table 1 and line 422) or C. hathewayi instead of C. hatheway (Table 1 and line 427).

Thanks for the deep revision of the manuscript. Spelling bacterial names has been modified.

Comment #5: Please use italics for the names of bacterial taxa (Table 1, Gut microbiota, penile microbiota)

We have checked all the bacteria taxa names and they have been italicized.

Comment #6: Please check the text to remove some typo and grammar errors. For example: line 290 “take” instead of “Take”; line 397 “axes” instead of “axis”; line 411 “been” instead of “be”; line 472 “these cases” instead of “the cases” and possibly in the other places.

Grammatical errors detected by the reviewer have been corrected. These have been highlighted for easy tracking.

Comment #7: In the literature section, position 121 there is not correct journal name. Should be “International Journal of Molecular Sciences”

We apologize for this error, the name of the journal has been modified to include “Int. J. Mol. Sci.”

Comment #8: In the section “3. The interplay between the different microbiotas” two sentences contain repeated information: lines 397-388 “The short-chain fatty acids (SCFAs), such as butyrate, propionate, acetate, or lactate are the best studied and the most prominent immunomodulatory metabolites [128].” and lines 390-391 “SCFAs are a by-product of fiber fermentation by certain intestinal microorganisms, being butyrate, propionate, acetate, and lactate the most representative ones [113,127]”. Perhaps in the second case would be better to list the bacterial species which produce SCFs?

At the reviewer's suggestion, the main SCFA-producing species have been included in the text using the phrase: “SCFAs are a by-product of fiber fermentation by certain intestinal microorganisms, being Roseburia intestinalis, Faecalibacterium prausnitzii, Eubacterium hallii, Bacteroides uniforms, Prevotella copri, Akkermansia muciniphila, Bifidobacterium spp. and Lactobacillus spp. the most important SCFAs producers [113,127]”

Finally, we would like to acknowledge the in-depth review carried out by the reviewer, taking into account the details that may have escaped the authors during the writing of the manuscript.

Due to the relatively little information concerning this topic, and even less experimental evidence, it is difficult for us to give specific mechanisms that explain the physiological or metabolic pathways behind the proposed interactions. Future investigations are expected to delve deeper into this matter. In the present manuscript, however, we tried to call the scientific community, and particularly those researchers that are expert in the field of the human microbiota, to devote attention to this topic that. While we recognize that our hypothesis can be risky, we think that we can not ignore the new findings.

We believe that this review is an opportunity to anticipate what we will see in the publications of the coming years where the information obtained from the sequencing of the microbiota of different niches can be integrated into predictive models with artificial intelligence, allowing a more inclusive approach than the one currently being carried out.

We hope that the changes made in this regard meet the expectations generated to the reviewer after reviewing the new version of the manuscript.

Reviewer 2 Report

The authors present a comprehensive overview of associations between human disease and alterations in microbial communities in different anatomical compartments in the body. The aim of the paper is to present evidence in support of the hypothesis that these microbial ecosystems are not independent but strongly intercommunicating, and therefore novel system approaches should be used to assess, study and potentially correct microbial dysfunctions.

The paper is well organised and covers various aspects of microbial dysbiosis. The association with human disease is well presented. The association between alterations in microbial communities belonging to different anatomical sites is also adequately described. However, one limitation of the review is that it does not adequately describe the mechanisms behind microbial intercommunication and whether disease-related phenomena may be the more likely mechanisms responsible for the selection of specific microbial species in different organs. The authors should give more visibility to this issue.

Author Response

REVIEWER #2

Comment #1: The authors present a comprehensive overview of associations between human disease and alterations in microbial communities in different anatomical compartments in the body. The aim of the paper is to present evidence in support of the hypothesis that these microbial ecosystems are not independent but strongly intercommunicating, and therefore novel system approaches should be used to assess, study and potentially correct microbial dysfunctions.

We greatly appreciate the comments of the reviewer who has captured the essence of the manuscript. It has not been an easy task to express the main idea that the authors have and translate it into a review. We appreciate the reviewer's comments because they have helped to better understand the main hypothesis of the manuscript.

Comment #2: The paper is well organised and covers various aspects of microbial dysbiosis. The association with human disease is well presented. The association between alterations in microbial communities belonging to different anatomical sites is also adequately described. However, one limitation of the review is that it does not adequately describe the mechanisms behind microbial intercommunication and whether disease-related phenomena may be the more likely mechanisms responsible for the selection of specific microbial species in different organs. The authors should give more visibility to this issue.

The manuscript attempted to formulate the hypothesis that the human microbiota could be considered as another organ system of the human body that behaves as a unique entity and have a strong impact on health. Since this hypothesis has never been studied in deep (due to the relatively little information concerning this topic, and even less experimental evidence) it is difficult to give specific mechanisms that explain the physiological or metabolic pathways behind the proposed interactions. Future investigations are expected to delve deeper into this matter. In the present manuscript, however, we tried to call the scientific community, and particularly those researchers that are expert in the field of the human microbiota, to devote attention to this topic that. While we recognize that our hypothesis can be risky, we think that we can not ignore the new findings.

Regarding the reviewer's comment, we have include the following paragraph in the Discussion section:

“Human microbiome analysis has been largely based on observation, with associations of disease phenotypes with particular microbiota constituents. However, one of the most controversial points in the study of the human microbiota is to establish whether the presence of a certain population of microorganisms is a cause or effect of the underlying disease and how this change can affect other niches where a specific microbiota resides. Different mechanisms can explain this connection, from metabolites (such as SCFA) to part of bacteria (such as extracellular bacterial vesicles) that migrate from different parts of the human body, to even the bacteria themselves that can cross epithelial barriers (such as the intestine epithelial cells) that lose their integrity in disease conditions (such as obesity,...). Extracellular bacterial vesicles have caught the attention of researchers [121, 122] as one of the mechanisms by which distant microorganisms could communicate, as it has been shown to occur with exosomes as an intercellular communication system in multi-cellular organisms.

Changes in the local microbiota occur in close contact with nearby cells, both host cells (with which there is a symbiotic or commensal relationship) and with nearby microorganisms with which they compete for the location and the nutrients in their environment. In this sense, the equilibrium that occurs is dynamic depending on multiple factors, both intrinsic (metabolism of the microorganisms present) and extrinsic (nutrients, pH conditions, oxygen pressure,...) that ultimately modulate the local microbiota present in a certain organ. In turn, the host cells are also influenced by the presence of a certain micro-biota and respond to it by adapting in a truly dynamic equilibrium that, when disrupted, is responsible for the development of a disease.”

We hope that the changes made in this regard meet the expectations generated to the reviewer after reviewing the new version of the manuscript.

We believe that this review is an opportunity to anticipate what we will see in the publications of the coming years where the information obtained from the sequencing of the microbiota of different niches can be integrated into predictive models with artificial intelligence, allowing a more inclusive approach than the one currently being carried out.